



**Ocean acidification and nutrient limitation synergistically reduce growth and photosynthetic performances of a green tide alga *Ulva linza***

Guang Gao[a,c], John Beardall[d], Menglin Bao[a], Can Wang[a], Wangwang Ren[b], Juntian Xu[a,b*]

[a]Marine Resources Development Institute of Jiangsu, Huaihai Institute of Technology, Lianyungang, 222005, China

[b]Co-Innovation Center of Jiangsu Marine Bio-industry Technology, Lianyungang 222005, China

[c]State Key Laboratory of Marine Environmental Science, Xiamen University, Xiamen 361005, China

[d]School of Biological Sciences, Monash University, Clayton, Victoria 3800, Australia

[*]Corresponding author, xjtlsx@126.com, Phone/Fax: +86 (0)518 8558 5003





**Abstract.** Large-scale green tides have been invading the coastal zones of the western
Yellow Sea annually since 2008. Meanwhile, oceans are becoming more acid due to
continuous absorption of anthropogenic carbon dioxide and intensive seaweed
cultivation in Chinese coastal areas is leading to severe regional nutrient limitation.
However, little is known the combined effects of global and local stressors on the
eco-physiology of bloom-forming algae. We cultured *Ulva linza* under two levels of
$pCO_2$ (400 and 1000 μatm) and four treatments of nutrient (nutrient repletion, N
limitation, P limitation, and N & P limitation) to investigate the physiological
responses of this green tide alga to the combination of ocean acidification and nutrient
limitation. For both sporelings and adult plants, elevated $pCO_2$ did not affect the
growth rate when cultured under nutrient replete conditions but reduced it under P
limitation; N or P limitation by themselves reduced growth rate. P limitation resulted
in a larger inhibition in growth for sporelings compared to adult plants. Sporelings
under P limitation did not reach the mature stage after 16 days of culture while those
under P repletion became mature by day 11. Elevated $pCO_2$ reduced net
photosynthetic rate for all nutrient treatments but increased nitrate reductase activity
and soluble protein content under P replete conditions. N or P limitation reduced
nitrate reductase activity and soluble protein content. These findings indicate that
ocean acidification and nutrient limitation would synergistically reduce the growth of
*Ulva* species and may thus hinder the occurrence of green tides in a future ocean
environment.
**Keywords:** green tide, growth, nitrate reductase, nutrient limitation, ocean





acidification, photosynthesis

**1 Introduction**

Seaweeds are a group of organisms that play a vital role in the function of coastal
ecosystems. They provide diverse habitats and breeding areas for a large number of
organisms including crustaceans, other invertebrates and fishes. In addition, in spite of
only occupying a small part of the world's oceans, seaweeds account for
approximately 10% of the total oceanic primary productivity due to their high
densities (Wiencke and Bischof, 2012). Consequently, they are of importance in
global carbon cycle and modulating climate change. In addition to high ecological
significance, seaweeds are also economically important. They have been widely used
in the industry for food, chemical products, pharmaceuticals, cosmetics, etc. (Wang et
al., 2017). The increasing demand has resulted in the fast development of seaweed
cultivation and successful cultivation has been carried out worldwide, particularly in
Asian countries. Among the diverse range of seaweeds, *Ulva*, a cosmopolitan genus in
green seaweeds, is common from tropical to polar areas, from fresh water to fully
saline environments due to its robustness in acclimating to a variety of salinity and
water temperature conditions (Gao et al., 2017b). Thanks to their strong capacity for
nutrient uptake and quick growth, *Ulva* is the only genus that causes green tides due
to massive growth (Smetacek and Zingone, 2013; Gao et al., 2017d). Green tides have
received increasing concerns globally due to their ecological and economic impacts.
Firstly, they can hinder shore-based activities by preventing small boats, swimmers
and tourists from accessing the sea due to their sheer physical mass. Furthermore,



nutrients could be re-released to the seawaters and toxic hydrogen sulphide ($H_2S$)
could be produced when thalli decompose, leading to highly eutrophic, anoxic
conditions and the spread of coastal dead zones (Smetacek and Zingone, 2013).
Atmospheric carbon dioxide ($CO_2$) has continuously increased from 278 to 407
parts per million since the industrial revolution (NOAA 2017). The global ocean has
absorbed around 30% of anthropogenic $CO_2$ emissions since 1750, leading to the
decrease of seawater pH termed ocean acidification (Gattuso et al., 2015). Ocean
acidification is not only changing the fundamental chemistry and physics in the
oceans but only imposing significant impacts on marine organisms (Mostofa, 2016).
In terms of marine algae, extensive studies have been conducted on phytoplankton
species (Mccarthy et al., 2012; Li et al., 2015; Cornwall et al., 2017a) or communities
(Gao et al., 2012; Eberlein et al., 2017; Gao et al., 2017c). Depending on experimental
conditions or species, the effects of ocean acidification on growth and photosynthesis
of phytoplankton could be positive (Mccarthy et al., 2012), neutral (Boelen et al.,
2011) or negative (Gao et al., 2012). Compared to phytoplankton, studies regarding
seaweeds are relatively few. Recently, however, there have been increasing concerns
about the responses of seaweeds, particularly *Ulva* species, to ocean acidification (Xu
and Gao, 2012; Rautenberger et al., 2015; Gao et al., 2016; Gao et al., 2017a). By
analyzing the literatures, it is found that the effects of ocean acidification on *Ulva*
species at different life stages are different. Ocean acidification generally increases
growth of *Ulva* species at early life stages (Xu and Gao, 2012; Gao et al., 2016) but
does not affect or even reduces growth of *Ulva* species at late life stages (Gao et al.,





2017a). A possible explanation for the differential effects of ocean acidification is that
higher $CO_2$ could induce more reproduction events for adult (Gao et al., 2017a).

Nutrients are crucial for growth and development of seaweeds. Nitrogen and

phosphorus, two key nutrient components for seaweeds, are commonly thought to be
limiting in natural seawater (Elser et al., 2007; Müller and Mitrovic, 2015).
Accordingly, enrichment of nitrogen and phosphorus generally stimulate growth of
seaweeds (Msuya and Neori, 2008; Luo et al., 2012; Xu et al., 2017). There are
studies indicating that N availability controls the biomass of seaweeds in temperate
coastal areas (Nixon and Pilson, 1983; Oviatt et al., 1995; Howarth et al., 2000) and P
limitation is the dominating factor for macroalgal growth in tropical latitudes
(Lapointe, 1997; Lapointe et al., 2010). However, phosphorus appears to play a more
important role in limiting the growth of *Ulva* sp. compared to nitrogen in a temperate
coastal area (Villares et al., 1999). In addition, Teichberg et al. (2010) investigated the
effects of nitrogen and phosphorus enrichment on growth of *U.* spp. at nine sites
across temperate and tropical areas. It was found that *Ulva*'s growth was controlled by
dissolved inorganic nitrogen (DIN) when ambient DIN levels were low and by
phosphorus when DIN levels were higher, irrespective of geographic or latitudinal
differences (Teichberg et al., 2010).

In addition to independent effects, the combined effects of ocean acidification and

nutrient on seaweeds have also been studied. Baydend et al. (2010) documented that
both ocean acidification and elevated nutrient concentration reduced the growth of
coralline crusts and the combination of the factors led to a further decrease in growth.





However, Xu et al. (2017) reported that ocean acidification and P enrichment did not
enhance the growth of brown seaweed *Sargasssum muticum* further applied together,
although either alone had a positive effect. The studies above indicate that the
combined effects of ocean acidification and eutrophication might be species-specific.

Until now, most studies regarding the effect of ocean acidification on seaweeds

have been based on nutrient replete conditions. In the field, the nutrient levels could
be limiting and this is particularly true in the areas of intensive seaweed cultivation.
For instance, the nitrogen and phosphorus concentrations in *Porphyra* cultivation
areas could be half of those in non-cultivation areas (He et al., 2008; Wu et al., 2015).
Little is known that how seaweeds grown under nutrient limited conditions respond to
ocean acidification. In addition, the young and adult thalli may have differential
responses to ocean acidification and nutrient limitation (Gao et al., 2017a). Therefore,
here we investigated the effects of ocean acidification and nutrient limitation on the
ecologically and commercially important seaweed *U. linza* at different life stages to
understand how *Ulva* species respond to the combination of global climate change
and local stressors.
**2 Materials and methods**
**2.1 Sample preparation and culture conditions**

To investigate whether life stage affects algal response to ocean acidification and

nutrient, both spores and adults of *U. linza* were used in this study. Fertile and
vegetative thalli (~5 cm) were collected from the coastal water of Lianyungang (119.3
$^{o}$E, 34.5 $^{o}$N), Jiangsu province, China. The fronds were put into in a cooling box (4–6



°C) and taken to the laboratory within 1 h. They were then rinsed with filtered (0.2 μm)
natural seawater to remove any sediment and small grazers. Spores, released from the
fertile thalli after being exposed to high light (600 μmol) for 12 h, were allowed to
settle and attach to glass slides in darkness for 12 h.
The settled spores were cultured in four nutrient regimes (higher nitrate and
higher phosphate (HNHP), lower nitrate and higher phosphate (LNHP), higher nitrate
and lower phosphate (HNLP), lower nitrate and lower phosphate (LNLP)) and two
$pCO_2$ (400 (LC) and 1000 (HC) μatm) to explore the interactive effects of ocean
acidification and nutrient limitation. The treatment of LCHNHP was set as the control.
Twenty volumes of natural seawater (30.7 μmol $L^{-1}$ N and 1.0 μmol $L^{-1}$ P) were
diluted with 80 volumes of artificial seawater without N or P to make a LNLP
medium (6.1 μmol $L^{-1}$ N and 0.2 μmol $L^{-1}$ P). The medium for P limitation (HNLP,
106.1μmol $L^{-1}$ N and 0.2 μmol $L^{-1}$ P) was made of LNLP medium plus 100 μmol N.
The medium for N limitation (LNHP, 6.1 μmol $L^{-1}$ N and 10.2 μmol $L^{-1}$ P) was made
of LNLP medium plus 10 μmol P. The N&P replete medium (HNHP, 106.1 μmol $L^{-1}$
N and 10.2 μmol $L^{-1}$ P) was made of LNLP medium plus 100 μmol N and 10 μmol P.
The 400 μatm $pCO_2$ level was maintained by bubbling ambient air, and the 1000 μatm
$pCO_2$ level was achieved using a $CO_2$ plant chamber (HP1000 GD, Wuhan Ruihua
Instrument & Equipment Ltd, China) with the variation of $CO_2$ less than 5% of the set
values. The incubation light intensity was 300 μmol photons $m^{-2} s^{-1}$, with a 12: 12
(light: dark) light period, and the incubation temperature was 20°C. The culture
conditions for adult *Ulva* were the same as for the spores. The cultures were carried




out in triplicates and lasted 16 days for spores and 9 days for adult thalli. The media
were renewed every two days.
**2.2 Measurement of growth**

The variations in sporeling length and adult fresh mass (FM) were recorded every

two days. The length of sporelings was measured by a microscope (Leica DM500,
Germany) with a micro ruler. The fresh mass of adults was determined by weighing
using a balance (BS 124S, Sartorius, Germany) after removing surface water by
gently blotting the thalli with tissue paper. The specific growth rate (SGR) was
estimated as follows: SGR (%) = $(\ln M_{t2} - \ln M_{t1}) / t \times 100$, where $M_{t1}$ is the initial
length for sporelings or initial fresh mass for adults; $M_{t2}$ is the length or fresh mass
after t days culture. Due to the tiny mass of sporelings, length rather than mass was
used to determine SGR for sporelings and only adult thalli were used for
measurements of the following aspects of physiological performance.
**2.3 Chlorophyll fluorescence assessment**

The relative electron transport rate (rETR) was measured using a pulse amplitude

modulation (PAM) fluorometer (PAM-2100, Walz, Germany). The measuring light
and actinic light was 0.01 and actinic light was set as the same as the growth light
(300 µmol photons $m^{-2}$ $s^{-1}$), respectively. The saturating pulse was set to 5, 000 µmol
photons $m^{-2}$ $s^{-1}$ (0.8 s). rETR (µmol $e^{-}$ $m^{-2}$ $s^{-1}$) = $(F_m' - F_t) / F_m' \times 0.5 \times PFD$, where
$F_m'$ is the maximal fluorescence levels from algae in the light. Ft is the fluorescence at
an excitation level. PFD is the actinic light density.
**2.4 Determination of photosynthesis**





The net photosynthetic rate of thalli was measured by a Clark-type oxygen
electrode (YSI model 5300A). Approximately 0.02 g of fresh weight algae with 8 ml
of media from the culture flask was transferred to the oxygen electrode cuvette, being
stirred. The conditions for temperature and light were set the same as those for growth.
The net photosynthetic rate was determined by the increase in the oxygen content in
the media over five minutes. The unit for net photosynthetic rate (NPR) was μmol $O_2$
$g^{-1}$ FM $h^{-1}$.
**2.5 Measurement of photosynthetic pigments**
Approximately 20 mg of fresh mass thalli was extracted in 5 mL methanol at $4^{o}$C
for 24 hours in darkness. Then the absorbance values of samples at 665 ($A_{665}$) and 652
($A_{652}$) were read with a UV/Visible spectrophotometer (Ultrospect 3300 pro,
Amersham Bioscience, Sweden). The content of Chl *a* and Chl *b* was determined as
follows:

Chl *a* (mg gFM$^{-1}$) = $(16.29 \times A_{665} - 8.54 \times A_{652}) \times V / (M \times 1000)$

Chl *b* (mg gFM$^{-1}$) = $(30.66 \times A_{652} - 13.58 \times A_{665}) \times V / (M \times 1000)$,

where V is the volume of methanol used and M is the mass of thalli used.
**2.6 Assessment of nitrate reductase activity**
Nitrate reductase activity of thalli was estimated based on a modified method of
Corzo and Niell (1991). The measurement was conducted during the local noon
period (13:00) since the activity of nitrate reductase usually displays circadian
periodicity with a maximum during the light period and a minimum in darkness
(Velasco and Whitaker 1989; Deng et al. 1991). Approximately 0.3 g (FM) of thalli





177 from each culture condition was incubated for 1 h at 20°C in darkness in the reaction

178 solution (10 mL), which contained 0.1 M phosphate buffer, 0.1% propanol (v/v), 50

179 mM $KNO_3$, 0.01 mM glucose, and 0.5 mM EDTA, with a pH of 8.0. The mixture was

180 flushed with pure $N_2$ gas (99.999%) for 2 minutes to obtain an anaerobic state before

181 the incubation. The concentration of nitrite produced was determined colorimetrically

182 at 540 nm (Xu et al., 2017). The NR activity was expressed as $\mu mol\ NO_2^-\ g^{-1}\ FM\ h^{-1}$.

183 **2.7 Estimation of soluble protein**

184 Approximately 0.2 g of FM thalli under each treatment at the end of the culture

185 period were ground in a mortar with extraction solution (0.1 mol $L^{-1}$ phosphate buffer,

186 pH 6.8) and then centrifuged for 10 minutes at 5, 000 $g$. Content of soluble protein

187 was estimated from the supernatant using the Bradford (1976) assay, with bovine

188 serum albumin as a standard.

189 **2.8 Statistical analysis**

190 The results in this study were expressed as means of replicates ± standard

191 deviation and the data were analyzed using the software SPSS v.22. The data under

192 every treatment conformed to a normal distribution (Shapiro-Wilk, $P > 0.05$) and the

193 variances could be considered equal (Levene's test, $P > 0.05$). Two-way multivariate

194 analysis of variance (MANOVA) was conducted to assess the effects of $pCO_2$ and

195 nutrient on seawater carbonate parameters. Repeated measures analysis of variance

196 (RM-ANOVA) was conducted to analyze the effects of culture time on length of

197 young and adult thalli, with Bonferroni for *post hoc* investigation. Two-way analysis

198 of variance (ANOVA) was conducted to assess the effects of $pCO_2$ and nutrient on



specific growth rate, net photosynthesis rate, rETR, Chl *a*, Chl *b*, soluble protein and
nitrate reductase activity. Tukey's honest significant difference (Tukey HSD) was
conducted for MNOVA and ANOVA *post hoc* investigation. Paired t-tests were used
to compare the differences in specific growth rate between young and adult thalli
under each treatment. A confidence interval of 95% was set for all tests.
**3 Results**
The carbonate system under each treatment was recorded (Table 1). Both $pCO_2$
and nutrient treatments had a significant effect on carbonate parameters (Table 2).
Elevated $pCO_2$ reduced pH and $CO_3^-$, increased DIC, $CO_2$ and $HCO_3^-$ but did not
affect TA (Tukey HSD, $P < 0.05$). P limitation (LP) increased $pCO_2$ and $CO_2$, and
reduced pH and $CO_3^{2-}$ (Tukey HSD, $P < 0.05$).
The length for both young and adult *U. linza* varied with culture time and the
patterns under different $pCO_2$ and/ or nutrient conditions were inconsistent (Fig. 1 &
Table 3). For example, the length gap between HP and LP increased with culture time
(Bonferroni, $P < 0.05$). It is worth noting that LP dramatically inhibited the
development of sporelings as the length under HP was 6,880−16,0290 μm while it
was only 137−250 μm under LP at the end of 16 days of culture.
Based on the initial and final length (young thalli) or mass (adult thalli), specific
growth rate was calculated (Fig. 2). Nutrient and $pCO_2$ interacted to affect the growth
of both young and adult *U. linza* (Table 4). Specifically, *post hoc* Tukey HSD
comparison ($P = 0.05$) showed that HC reduced growth at LP but did not affect it at
HP, suggesting an interactive effect between P and C. Nutrient supply had an effect on



growth but the patterns between young and adult thalli were different (Table 4). For
young thalli, *post hoc* Tukey HSD comparison ($P = 0.05$) showed that N limitation
did not reduce growth, P limitation dramatically reduced the growth and the
combination of N and P limitation did not lead to a further decrease regardless of
$pCO_2$ conditions. For adult thalli, either N or P limitation reduced growth and the
combination of these nutrient limitations resulted in a further decrease (Tukey HSD, $P$
$< 0.05$). In addition, young thalli had higher growth rates under each condition
compared to adult plants (Paired t-test, $P < 0.05$).

The effects of $pCO_2$ and nutrients on the net photosynthetic rate of adult thalli

were also investigated (Fig. 3). Both $pCO_2$ and nutrient had a significant effect on net
photosynthetic rate (Table 5) and HC reduced NPR under each nutrient condition
(Tukey HSD, $P < 0.05$). In terms of the effect of nutrients, LN or LP alone decreased
NPR and the combination of LN and LP led to a further decrease under LC (Tukey
HSD, $P < 0.05$). Under HC, *post hoc* Tukey HSD comparison ($P = 0.05$) showed that
both LN and LP reduced NPR but the combination of LN and LP did not decrease
NPR further.

To understand the photosynthetic performance of *U. linza* under various $pCO_2$

and nutrient conditions, relative electron transport rate (rETR) in PSII at 300 μmol
photons m$^{-2}$ s$^{-1}$ was measured (Fig. 4). $pCO_2$ had an interactive effect with nutrient
and each factor had a main effect (Table 5). Specifically speaking, HC reduced rETR
under LP but did not change it under HP. Regardless of $pCO_2$ levels, N limitation
reduced rETR (Tukey HSD, $P < 0.05$), P limitation had a larger negative effect and



the combination of LN and LP resulted in the lowest rETR values (Tukey HSD, $P <$

0.05).

Changes in photosynthetic pigments are shown in Fig. 5. Both $pCO_2$ and nutrient

had an effect on the content of Chl $a$ and Chl $b$ (Table 4) but slight differences
between Chl $a$ and Chl $b$ were found after post hoc Tukey HSD ($P < 0.05$) tests had
been conducted. Under LC, either N or P limitation reduced Chl $a$ content, with P
limitation having a larger effect. LNLP decreased Chl $a$ content further. Under HC,
LN or LP reduced Chl $a$ but the combination of limiting nutrients did not lead to a
further decrease. As far as Chl $b$ is concerned, either LN or LP decreased Chl $b$, with
LP having a larger effect under LC. The combination of LN and LP did not lead to a
further decrease compared with LP.

To investigate the effects of $pCO_2$ and nutrient on nitrogen acquisition, nitrate

reductase activity (RNA) in adult $U. linza$ grown under various conditions was
measured (Fig. 6). Both $pCO_2$ and nutrient affected NRA and they had an interactive
effect (Table 7). Under LC, $post hoc$ Tukey HSD comparison ($P = 0.05$) showed that
either N or P limitation reduced NRA but the combination of them did not result in a
further decrease. Under HC, N limitation and P limitation reduced NRA by 22.8% and
37.7% respectively and the combination of them increased NRA by 45.6%. In
addition, HC did not affect NRA under LNLP (Tukey HSD, $P > 0.05$) but increased it
when N or P was replete and nitrate reductase had the highest activity ($11.9 \pm 0.7$
$\mu$mol $NO_2^-$ $g^{-1}$ FM $h^{-1}$) under HCHNHP condition (Tukey HSD, $P < 0.05$).

The content of soluble protein was assayed to investigate nitrogen assimilation of





*U. linza* under various $pCO_2$ and nutrient conditions (Fig. 7). Both $pCO_2$ and nutrient
levels affected the content of soluble protein and showed interactive effects (Table 7).
*Post hoc* Tukey HSD comparison ($P = 0.05$) showed that HC did not affect the
content of soluble protein under LP but increased it under HP. Under LC, separate N
or P limitation and their combination showed a similar negative effect on soluble
protein content. Under HC, P limitation had a larger inhibition effect on soluble
protein content compared to N limitation (Tukey HSD, $P < 0.05$) and the combination
of N and P limitation did not lead to a further decrease in soluble protein content.
**4 Discussions**
**4.1 Differential response of young and adult *Ulva***
Compared to adult plants, young *Ulva* grew much faster regardless of culture
conditions. This trend was also found in *U. rigida* (Gao et al., 2017a). The noticeable
difference in growth rate between young and adult *Ulva* could be attributed to cell
differentiation. In the early life history of *Ulva*, cell division proceeds fast as all cells
are of the same type, developing from one single cell. Thereafter, cells differentiate
into two types: rhizoidal cells in the basal part and blade cells in the marginal part
(Gao et al., 2017b). Differences in cell size and photosynthetic pigments between
these two cell types result in unequal growth in the thallus; the growth of rhizoidal
cells is much slower than in blade cells (Han et al., 2003; Lüning et al., 2008), which
slows down the total growth of the thallus as it ages.
Lower P levels strongly inhibited the growth of both young and adult plants in this
study and the inhibitory effect was particularly significant for young plants. Gao et al.



(2017b) has reported that *U. rigida* becomes mature when the thalli reach a length of
around 1.5 cm and we also found a similar phenomenon in *U. linza*. Young plants
grown under P limitation were far away the mature stage even after 16 days of culture
while the plants grown under P repletion reached a mature stage by day 11 and the
length could be up to 16 cm by day 16. This finding supports the significant role of P
in development of *U. linza*. Phosphorus (P) is an essential element for seaweeds, in
the form of nucleic acids, phospholipids, ATP and ADP, but little is known regarding
the effect of P on development of seaweed. Our findings indicate that P limitation
may terminate the development of young *Ulva* and cause it to remain in the immature
stage. The separate addition of N did not change the growth rate of young plants but
increased the growth rate of adult plants, suggesting that adult plants could be more
resilient to P limitation compared to young plants.
**4.2 Photosynthetic response to OA and nutrients**
HC was shown in the present study to decrease the Chl *a* and Chl *b* contents of *U.*
*linza*. High $CO_2$ commonly down-regulates algal $CO_2$ concentrating mechanisms
(CCMs), suggesting less energy is required to drive CCMs (Gao et al., 2012; Raven et
al., 2012; Cornwall et al., 2017b; Raven et al., 2017). This may lead to decreased
synthesis of pigment for energy capture. This phenomenon of `pigment economy' has
also been found in our previous studies regarding *Ulva* species (Xu and Gao, 2012;
Gao et al., 2016). Deficiency in N or/and P supply also reduced pigment content in
this study. Nitrogen is a major component of Chl *a*. Although P is a non-constituent
element in Chl *a*, higher P supply may stimulate the activity of Chl *a* synthesis-related



enzymes (Xu et al., 2017). Accordingly, in this study nutrient (N & P) enrichment
enhanced the synthesis of Chl *a*. This is consistent with other findings in *Ulva* species
(Gordillo et al., 2001; Figueroa et al., 2009) and other macroalgae (Xu et al., 2017).

HC decreased net photosynthetic rate in *U. linza* in the present study. This could be

due to the decrease of photosynthetic pigment in thalli grown under HC. Meanwhile,
the down-regulation of CCMs in thalli grown under HC might have reduced the
intracellular $CO_2$ availability and have contributed to the lower net photosynthetic rate.
An ocean acidification-induced decrease of net photosynthetic rate was also
documented in *U. prolifera* (Xu and Gao, 2012). In terms of the effects of nutrient, N
limitation reduced net photosynthetic rate in *U. linza* and P limitation resulted in a
further decrease. The negative effects of N and P limitation on algal photosynthetic
rate have been extensively reported (Longstaff et al., 2002; Kang and Chung, 2017;
Xu et al., 2017), indicating the important role of N and P in algal photosynthesis. In
addition to the separate effects of $pCO_2$ or nutrient, these factors also interplay on
photosynthetic performances of *U. linza*. For instance, HC reduced rETR under LP
but did not affect it under HP, suggesting that P enrichment could offset the negative
effect of ocean acidification.
**4.3 N assimilation under OA and nutrient limitation**

Contrary to C assimilation, HC did not affect the content of soluble protein under

LP and even increased it when the P level was sufficient. The increased protein
synthesis under HC could be put down to the stimulation of NRA activity under HC.
Gordillo et al. (2001) proposed that the positive effect of HC on N assimilation may





be due to a direct action on synthesis of NR rather than the physiological
consequences of C metabolism as occurs in higher plants. Our results support this
hypothesis because HC increased NRA in thalli grown under HP in this study. P is
considered to be playing a critical role in enzyme synthesis and may interact with $CO_2$
to promote the synthesis and activity of nitrate reductase.
**4.4 Interactive effects of OA and nutrient limitation on Growth**
HC did not affect the growth of thalli when P was replete in the medium. Since HC
reduced photosynthesis rate but increased NRA and protein synthesis, the lack of
effect of HC may be an integrated outcome of C and N assimilation. This finding is
different from our previous studies in which HC increased the growth rate of *U. linza*
(Gao et al., 2016) and *U. prolifera* (Gao et al., 2017d). The possible reason causing
this divergence might be due to the different light intensities that were used in the
various studies. For the previous studies, a lower light intensity of 100 μmol photons
$m^{-2} s^{-1}$ was used for algal culture while a higher light intensity of 300 μmol photons
$m^{-2} s^{-1}$ was used in the present work. Ocean acidification could interact with light
intensity to affect algal growth. It has commonly been reported that ocean
acidification can increase algal photosynthesis/growth at lower light intensity and
inhibit photosynthesis/growth at higher light intensity (Gao et al., 2012; Xu and Gao,
2012; Gao et al., 2016), with inversion points of PAR around 160, 125 and 178 μmol
photons $m^{-2} s^{-1}$ for *Phaeodactylum tricornutum*, *Thalassiosira pseudonana* and
*Skeletonema costatum* respectively (Gao et al., 2012). It seems that *U. linza* has a
higher inversion point compared to diatoms.





Furthermore, HC reduced growth of *U. linza* when P was limited. In addition to the
increased $CO_2$ supply, ocean acidification also reduces the pH of seawater, which has
been deemed as a stressor disturbing the acid-base balance both at the cell surface and
within cells and affecting algal photosynthetic performance (Flynn et al., 2012; Gao et
al., 2017d). Increased $CO_2$ and decreased pH also reduced rETR and net
photosynthetic rate of *U. linza* in the present study. Xu et al. (2017) proposed that
algae could synthesize HC transport-related proteins to combat that disturbance.
Under P limitation conditions, such protein synthesis could be limited, which may
lead to the decreased rETR and net photosynthetic rate and thus to decreased growth
observed in the present study. Until now, most studies of ocean acidification on
seaweed have been conducted under nutrient replete conditions. The present study
thus demonstrates the contrasting effect of ocean acidification under nutrient deplete
conditions.
**4.5 Differential effects of N and P limitation**
In the present study, compared to N limitation, P limitation seems to have a larger
negative effect on physiological performances in *U. linza*. In other words, the addition
of P resulted in a larger stimulating effect compared to N addition. Which one (N or P)
is the nutrient most likely to limit marine primary productivity has been a
controversial issue until now (Elser et al., 2007; Teichberg et al., 2010; Müller and
Mitrovic, 2015). It has been proposed that the occurrence of N or P limitation depends
on the difference in N:P ratio between in algal tissue and in seawater; when the ratio
of N:P in algal tissue is higher than in seawater N limitation is indicated and the



opposite is considered as P limitation (Harrison and Hurd, 2001). The ratio of N:P in
tissue of *U. linza* grown in the field has not been documented and the mean value of
N:P throughout a year is 15.4 for *U. prolifera* and 22.3 for *U. fenestrate* (Wheeler and
Björnsäter, 1992). The ratio of N:P in natural seawater where the samples were
collected in the present study is 30.7:1, which is higher than the N:P ratio in the
reported *Ulva* species. This suggests the existence of P limitation for *U. linza*, which
could explain the larger stimulating effect with P addition.
In recent decades, P limitation has been suggested to commonly occur in coastal
waters due to more effective P removal from industrial and domestic wastewater
during de-eutrophication processes (Grizzetti et al., 2012). For instance, the ratio of
dissolved inorganic N:P could be as high as 375:1 in nearshore waters of the North
Sea, resulting in severe P limitation for algal growth (Burson et al., 2016).
**5 Conclusions**
With the continuous emission of $CO_2$, the trend of ocean acidification will
continue through this century (Gattuso et al., 2015). Meanwhile, nutrient limitation
would occur in coastal waters as a consequence of efforts on de-europhication.
Measures to reduce eutrophication have often led to a more effective decline of
phosphorus (P) than nitrogen (N) concentrations (Burson et al., 2016). In addition,
intensive seaweed culture in coastal areas can also lead to noticeable decreases in N
and P (He et al., 2008; Wu et al., 2015). Our study demonstrates that ocean
acidification and nutrient limitation would synergistically inhibit development and
growth of *Ulva* species. This may hinder the occurrence of green tides in future ocean.



In addition, it has been reported that fast-growing species require high nutrient inputs
to sustain growth, while slow-growing species are better adapted to nutrient limiting
conditions (Gordillo, 2012). The decrease in nutrient level may result in a shift in
seaweed community composition in the future ocean environment. Studies on other
seaweeds are needed to have a comprehensive understanding in terms of the
combined effects of global and local stressors on seaweed communities.
**Acknowledgements**

This study was supported by the National Natural Science Foundation of China

(No. 41476097), the Natural Science Foundation of Jiangsu Province (No.
BK20161295), the Science and Technology Bureau of Lianyungang (SH1606), the
Jiangsu Planned Projects for Postdoctoral Research Funds (1701003A), the Science
Foundation of Huaihai Institute of Technology (Z2016007), and the Priority Academic
Program Development of Jiangsu Higher Education Institutions of China.

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




**Table 1.** Parameters of the seawater carbonate system in different cultures. LC, 400 μatm; HC, 1000 μatm; LN, 6.1 μmol L$^{-1}$; LP, 0.2 μmol L$^{-1}$; HN, 106.1 μmol L$^{-1}$; HP, 10.2 μmol L$^{-1}$. DIC = dissolved inorganic carbon, TA = total alkalinity. Data are the means ± SD (n = 3).

| Treatment | $p$CO$_2$ (μatm) | pH | DIC (μmol kg$^{-1}$) | CO$_2$ (μmol kg$^{-1}$) | HCO$_3^-$ (μmol kg$^{-1}$) | CO$_3^{2-}$ (μmol kg$^{-1}$) | TA (μmol kg$^{-1}$) |
|---|---|---|---|---|---|---|---|
| LCHNHP | 373.8±6.3 | 8.21±0.01 | 1991.9±49.4 | 12.3±0.2 | 1805.3±43.3 | 174.2±6.1 | 2243.7±55.8 |
| LCLNHP | 381.3±2.9 | 8.20±0.01 | 2015.1±50.4 | 12.6±0.1 | 1827.4±42.0 | 175.1±8.5 | 2267.0±60.5 |
| LCHNLP | 400.4±9.0 | 8.19±0.01 | 2029.9±50.1 | 13.2±0.3 | 1846.4±43.1 | 170.2±7.7 | 2262.7±58.2 |
| LCLNLP | 397.4±8.4 | 8.18±0.01 | 1998.2±39.0 | 13.1±0.3 | 1818.7±35.1 | 166.3±4.3 | 2226.7±42.9 |
| HCHNHP | 929.1±24.8 | 7.86±0.01 | 2154.6±52.5 | 30.7±0.8 | 2034.9±49.5 | 89.1±2.6 | 2263.3±54.2 |
| HCLNHP | 958.8±5.2 | 7.85±0.01 | 2155.0±42.0 | 31.6±0.2 | 2036.8±38.5 | 86.5±3.6 | 2259.0±46.5 |
| HCHNLP | 976.1±10.9 | 7.84±0.01 | 2159.3±38.7 | 32.2±0.4 | 2041.7±36.1 | 85.4±2.5 | 2250.7±41.3 |
| HCLNLP | 1020.2±51.8 | 7.82±0.03 | 2117.9±31.0 | 33.7±1.7 | 2005.3±28.0 | 78.9±5.2 | 2199.3±37.4 |





**Table 2.** Two-way multivariate analysis of variance for the effects of of $pCO_2$ and nutrient on on pH, dissolved inorganic carbon (DIC), $HCO_3^-$, $CO_3^{2-}$, $CO_2$, total alkalinity (TA) in the seawater. $pCO_2$*nutrient means the interactive effect of $pCO_2$ and nutrient, df means degree of freedom, F means the value of F statistic, and Sig. means *p*-value.

| Source | df | pH | | DIC | | $HCO_3^-$ | | $CO_3^{2-}$ | | $CO_2$ | | TA | |
|---|---|---|---|---|---|---|---|---|---|---|---|---|---|
| | | F | Sig. | F | Sig. | F | Sig. | F | Sig. | F | Sig. | F | Sig. |
| $pCO_2$ | 1 | 5237.765 | <0.001 | 57.132 | <0.001 | 158.536 | <0.001 | 1504.349 | <0.001 | 4486.773 | <0.001 | 0.114 | 0.740 |
| Nutrient | 3 | 9.765 | 0.001 | 0.747 | 0.540 | 0.741 | 0.543 | 3.336 | 0.046 | 7.999 | 0.002 | 1.225 | 0.333 |
| $pCO_2$*Nutrient | 3 | 1.294 | 0.311 | 0.256 | 0.856 | 0.332 | 0.802 | 0.162 | 0.921 | 2.683 | 0.082 | 0.228 | 0.876 |
| Error | 16 | | | | | | | | | | | | |





**Table 3.** Repeated analysis of variance for the effects of culture time on length changes of young and adult *U. linza* grown under various $pCO_2$ and nutrient conditions. Time*$pCO_2$ means the interactive effect of Time and $pCO_2$, Time* nutrient means the interactive effect of Time and nutrient, and Time*$pCO_2$*nutrient means the interactive effect of Time, $pCO_2$ and nutrient. df means degree of freedom, F means the value of F statistic, and Sig. means *p*-value.

| Source | Growth of young *U. linza* | | | Growth of adult *U. linza* | | |
|---|---|---|---|---|---|---|
| | df | F | Sig. | df | F | Sig. |
| Time | 8 | 1153.328 | <0.001 | 4 | 571.769 | <0.001 |
| Time*$pCO_2$ | 8 | 23.582 | <0.001 | 4 | 3.158 | 0.020 |
| Time*Nutrient | 24 | 457.170 | <0.001 | 12 | 28.505 | <0.001 |
| Time*$pCO_2$*nutrient | 24 | 10.585 | <0.001 | 12 | 0.689 | 0.756 |
| Error | 128 | | | 64 | | |



**Table 4.** Two-way analysis of variance for the effects of $pCO_2$ and nutrient on relative growth rate of *U. linza*. $pCO_2$*nutrient means the interactive effect of $pCO_2$ and nutrient, df means degree of freedom, F means the value of F statistic, and Sig. means *p*-value.

| Source | Growth of young *U. linza* | | | Growth of adult *U. linza* | | |
|---|---|---|---|---|---|---|
| | df | F | Sig. | df | F | Sig. |
| $pCO_2$ | 1 | 115.297 | <0.001 | 1 | 20.039 | <0.001 |
| Nutrient | 3 | 12678.566 | <0.001 | 3 | 307.073 | <0.001 |
| $pCO_2$*nutrient | 3 | 22.905 | <0.001 | 3 | 1.723 | 0.011 |
| Error | 16 | | | 16 | | |





**Table 5.** Two-way analysis of variance for the effects of $pCO_2$ and nutrient on net photosynthetic rate and rETR of *U. linza.* $pCO_2$*nutrient means the interactive effect of $pCO_2$ and nutrient, df means degree of freedom, F means the value of F statistic, and Sig. means *p*-value.

| | Net photosynthetic rate | | | rETR | | |
|---|---|---|---|---|---|---|
| Source | df | F | Sig. | df | F | Sig. |
| $pCO_2$ | 1 | 35.096 | <0.001 | 1 | 14.592 | 0.002 |
| Nutrient | 3 | 493.992 | <0.001 | 3 | 135.690 | <0.001 |
| $pCO_2$*nutrient | 3 | 2.619 | 0.087 | 3 | 5.023 | 0.012 |
| Error | 16 | | | 16 | | |





**Table 6.** Two-way analysis of variance for the effects of $pCO_2$ and nutrient on content of Chl $a$ and Chl $b$ in *U. linza*. $pCO_2$*nutrient means the interactive effect of $pCO_2$ and nutrient, df means degree of freedom, F means the value of F statistic, and Sig. means $p$-value.

| Source | Chl $a$ | | | Chl $b$ | | |
|---|---|---|---|---|---|---|
| | df | F | Sig. | df | F | Sig. |
| $pCO_2$ | 1 | 85.900 | <0.001 | 1 | 71.600 | <0.001 |
| Nutrient | 3 | 217.334 | <0.001 | 3 | 104.483 | <0.001 |
| $pCO_2$*nutrient | 3 | 2.440 | 0.102 | 3 | 2.005 | 0.154 |
| Error | 16 | | | 16 | | |





**Table 7.** Two-way analysis of variance for the effects of $pCO_2$ and nutrient on nitrate

reductase activity and soluble protein of *U. linza.* $pCO_2$*nutrient means the interactive

effect of $pCO_2$ and nutrient, df means degree of freedom, F means the value of F

statistic, and Sig. means *p*-value.

| Source | Nitrate reductase activity | | | Soluble protein | | |
|---|---|---|---|---|---|---|
| | df | F | Sig. | df | F | Sig. |
| $pCO_2$ | 1 | 38.271 | <0.001 | 1 | 30.212 | <0.001 |
| Nutrient | 3 | 100.487 | <0.001 | 3 | 106.523 | <0.001 |
| $pCO_2$*nutrient | 3 | 6.246 | 0.005 | 3 | 11.295 | <0.001 |
| Error | 16 | | | 16 | | |



**Figure legends**

**Fig. 1.** Length changes of young (a) and adult (b) *U. linza* grown under various conditions during the culture periods. LC, 400 μatm; HC, 1000 μatm; LN, 6.1 μmol L$^{-1}$; LP, 0.2 μmol L$^{-1}$; HN, 106.1 μmol L$^{-1}$; HP, 10.2 μmol L$^{-1}$.

**Fig. 2.** Specific growth rate (% d$^{-1}$) of young (a) and adult (b) *U. linza* grown under various conditions. The specific growth rate for young and adult thalli were calculated based on the initial and final length (for young thalli over a 16-day culture) or mass (for adult thalli over a 16-day). LC, 400 μatm; HC, 1000 μatm; LN, 6.1 μmol L$^{-1}$; LP, 0.2 μmol L$^{-1}$; HN, 106.1 μmol L$^{-1}$; HP, 10.2 μmol L$^{-1}$. Different letters (low-case for LC and capital for HC) above the bars represent significant differences ($P < 0.05$) among nutrient treatments while horizontal bars represent significant differences ($P < 0.05$) between LC and HC within a nutrient treatment.

**Fig. 3.** Net photosynthetic rate of adult *U. linza* grown under various conditions. LC, 400 μatm; HC, 1000 μatm; LN, 6.1 μmol L$^{-1}$; LP, 0.2 μmol L$^{-1}$; HN, 106.1 μmol L$^{-1}$; HP, 10.2 μmol L$^{-1}$. Different letters (low-case for LC and capital for HC) above bars represent significant differences *($P < 0.05$)* among nutrient treatments while horizontal bars represent significant differences ($P < 0.05$) between LC and HC within a nutrient treatment.

**Fig. 4.** Relative electron transport rate (rETR) of adult *U. linza* grown under various conditions. LC, 400 μatm; HC, 1000 μatm; LN, 6.1 μmol L$^{-1}$; LP, 0.2 μmol L$^{-1}$; HN, 106.1 μmol L$^{-1}$; HP, 10.2 μmol L$^{-1}$. Different letters above bars (low-case for LC and capital for HC) represent significant differences ($P < 0.05$) among nutrient treatments





while horizontal bars represent significant differences ($P < 0.05$) between LC and HC

within a nutrient treatment.

**Fig. 5.** Content of Chl *a* (a) and Chl *b* (b) in adult *U. linza* grown under various

conditions. LC, 400 μatm; HC, 1000 μatm; LN, 6.1 μmol L$^{-1}$; LP, 0.2 μmol L$^{-1}$; HN,

106.1 μmol L$^{-1}$; HP, 10.2 μmol L$^{-1}$. Different letters (low-case for LC and capital for

HC) above bars represent significant differences ($P < 0.05$) among nutrient treatments

while horizontal bars represent significant differences ($P < 0.05$) between LC and HC

within a nutrient treatment.

**Fig. 6.** Nitrate reductase activity (NRA) in adult *U. linza* grown under various

conditions. LC, 400 μatm; HC, 1000 μatm; LN, 6.1 μmol L$^{-1}$; LP, 0.2 μmol L$^{-1}$; HN,

106.1 μmol L$^{-1}$; HP, 10.2 μmol L$^{-1}$. Different letters (low-case for LC and capital for

HC) above bars represent significant differences ($P < 0.05$) among nutrient treatments

while horizontal bars represent significant differences ($P < 0.05$) between LC and HC

within a nutrient treatment.

**Fig. 7.** Content of soluble protein in adult *U. linza* grown under various conditions.

LC, 400 μatm; HC, 1000 μatm; LN, 6.1 μmol L$^{-1}$; LP, 0.2 μmol L$^{-1}$; HN, 106.1 μmol

L$^{-1}$; HP, 10.2 μmol L$^{-1}$. Different letters (low-case for LC and capital for HC) above

bars represent significant differences ($P < 0.05$) among nutrient treatments while

horizontal bars represent significant differences ($P < 0.05$) between LC and HC

within a nutrient treatment.





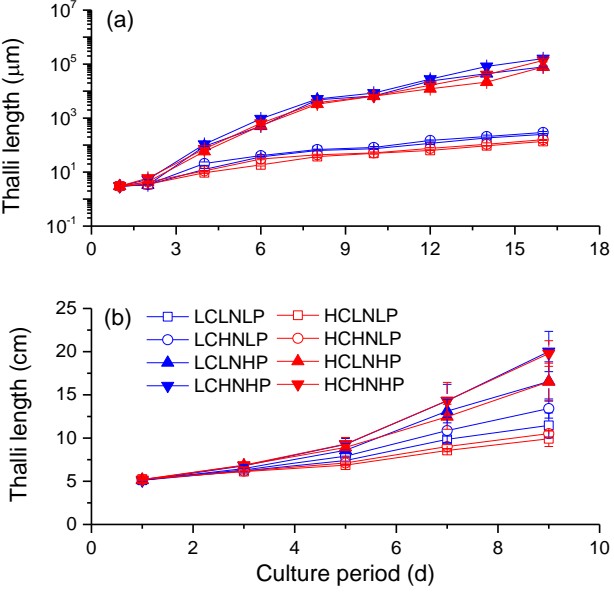

Fig. 1



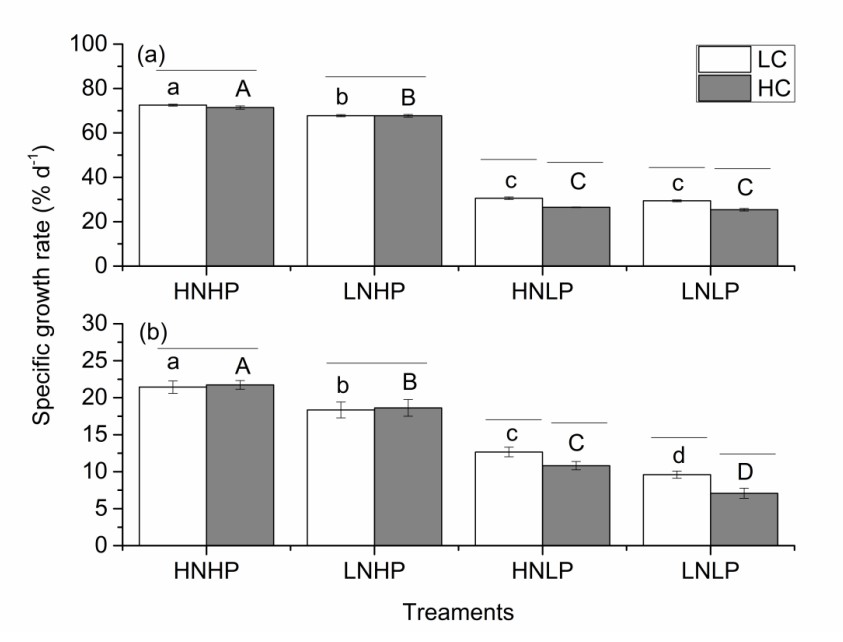

Fig. 2




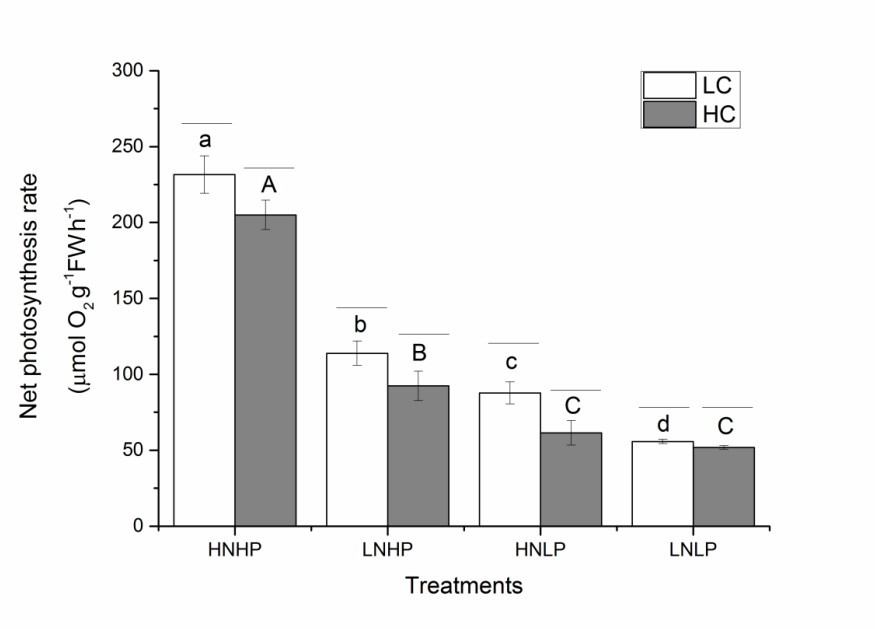

Fig. 3




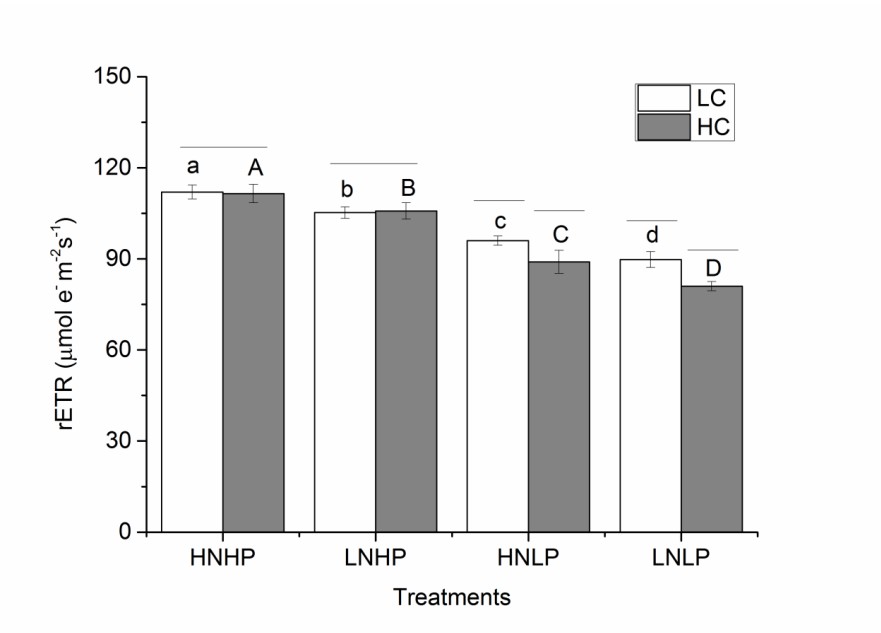

Fig. 4





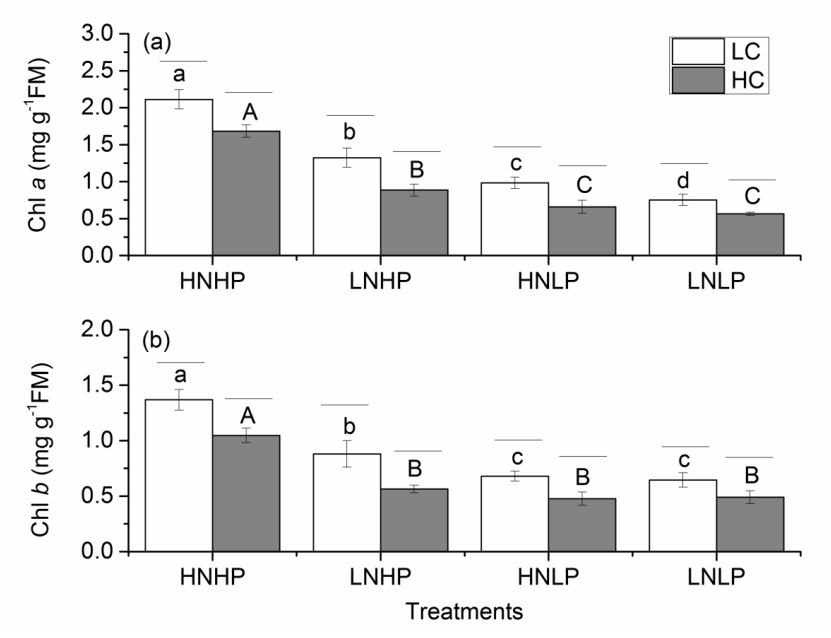

Fig.5



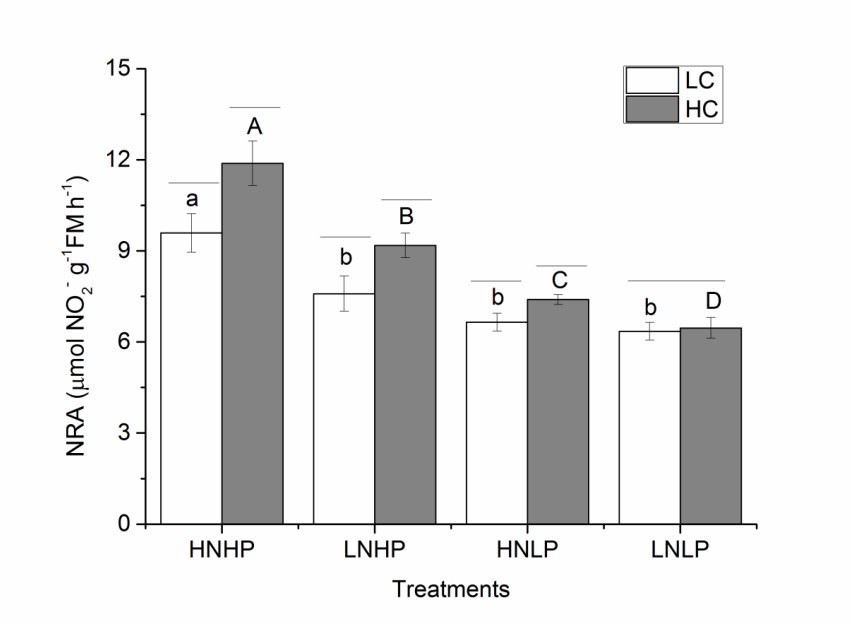

Fig. 6



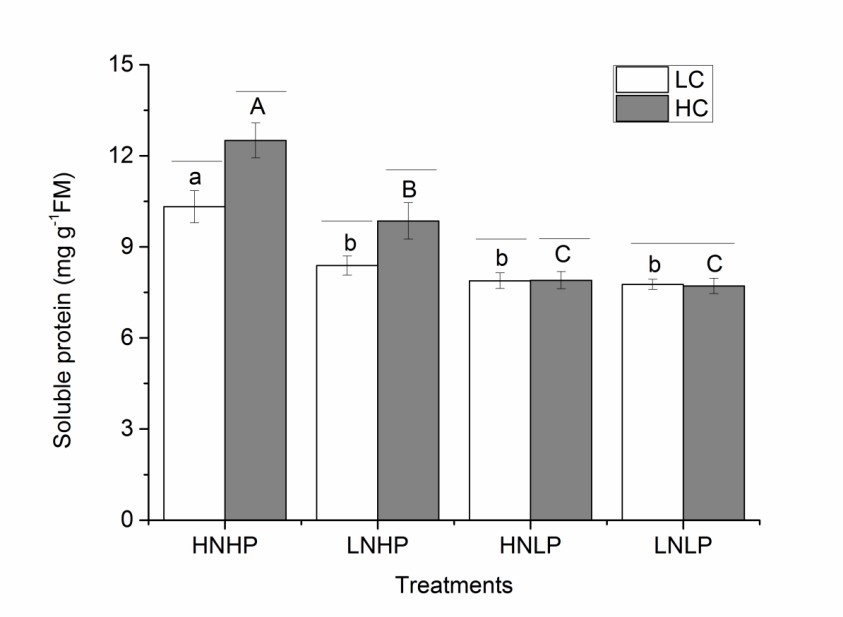

Fig. 7