# Peer review of "Ocean acidification and nutrient limitation synergistically reduce growth and photosynthetic performances of a green tide alga *Ulva linza"

_Biogeosciences, 2018_

## Short Comment (SC1) · 22 Feb 2018

The authors investigated the combined effects of ocean acidification and nutrient limitation on physiological performances, including growth, photosynthetic oxygen evolution, PSII fluorescence parameters, nitrogen assimilation, in a green tide alga, and found that ocean acidification did not affect growth and photosynthesis under the nutrient replete condition but reduced them when nutrient was limited. Nitrogen assimilation was stimulated by ocean acidification when nutrient was replete. The experiments were reasonably performed and the results were clearly presented. This study is of interest, indicating the interactive effects of global and local stressors on a green tide alga. But

there are still some points to be revised before it could be published in Biogeosciences. Major revisions 1. Why were different cultivation periods used for sporeling and adult thalli? Are these periods enough for algal acclimation to ocean acidification? 2. Please clarify the culture density used in this study and to what extent pH fluctuated during the culture period. How to maintain a stable pH in the cultures? 3. Why was the light density of 300 photons m-2 s-1 used for the cultures since lower levels were used for the previous studies as mentioned in the text. Is the one used in this study close to ambient sunlight? Minor revisions Line 113 change $\mu$mol to $\mu$mol photons m-2 s-1 Line 123 add a space after 106.1 Line 156 change weight to mass Line 329 delete activity and be consistent for using NRA or NR activity throughout the text. Figure 3 change FW to FM in Y axes legend Figure 7 I doubt there is a significant difference between HC and LC for the treatment of HNLP

---

## Referee Comment (RC1) · Anonymous Referee #2 · 2 Mar 2018

Journal: BG Title: Ocean acidification and nutrient limitation synergistically reduce growth and photosynthetic performances of a green tide alga Ulva linza Author(s): Guang Gao et al. MS No.: bg-2018-1

Comments: The authors investigated the combined effects of ocean acidification and nutrient limitation on physiological performances, including growth, photosynthetic oxygen evolution, PSII fluorescence parameters, nitrogen assimilation, in a green tide alga, and found that ocean acidification did not affect growth and photosynthesis under the nutrient replete condition but reduced them when nutrient was limited. Nitrogen assimilation was stimulated by ocean acidification when nutrient was replete. The experiments were reasonably performed and the results were clearly presented. This study is of interest, indicating the interactive effects of global and local stressors on a green tide alga. But there are still some points to be revised before it could be published in Biogeosciences. Special revisions 1. Why were different cultivation periods used for sporeling and adult thalli? Are these periods enough for algal acclimation to ocean acidification? 2. Please clarify the culture density used in this study and to what extent pH fluctuated during the culture period. How to maintain a stable pH in the cultures? 3. Why was the light density of 300 photons m-2 s-1 used for the cultures since lower levels were used for the previous studies as mentioned in the text. Is the one used in this study close to ambient sunlight? Minor revisions Line 113 change $\mu$mol to $\mu$mol photons m-2 s-1 Line 123 add a space after 106.1 Line 156 change weight to mass Line 329 delete activity and be consistent for using NRA or NR activity throughout the text. Figure 3 change FW to FM in Y axes legend Figure 7 I doubt there is a significant difference between HC and LC for the treatment of HNLP

---

## Referee Comment (RC2) · Anonymous Referee #3 · 27 Mar 2018

This paper reports results from an interesting study aiming to test the effects of ocean acidification and nutrients limitation on Ulva. The study is pretty straightforward: adult and juvenile algae were exposed to different conditions of CO2 and nutrients and their physiological response was investigated. While this study is rather "classical", the originality comes from the use of nutrient limitation, while most studies have used so far nutrients addition. The results are rather interesting and demonstrate that the interaction between pCO2 and nutrient limitations are not straightforward. I find the discussion a bit complex and hard to read given the quantity of physiological parameters discussed. It might be worth considering adding a figure that would summarize all the results. Maybe a schematic representing the physiological impact of nutrients and

carbon could be added.

I have listed below some specific comments.

Abstract: indicate the duration of the experiment

L55: Wrong reference for Cornwall et al. 2017, they looked at coralline algae not phytoplankton.

L63-64: reformulate this sentence

L119: "LCHNHP" is a bit hard to read/understand but I guess it's not really used later on.

L130: How does this light level compare to in situ?

L132: What was the size of the tanks? Did you use any pumps, etc , to create water motion? This is critical as it could affect the capacity of the organisms to uptake nutrients.

L133: Any reason to have chosen these durations? 9 days is rather short.

L156: What were those fragments? Just a piece of algae? I always have problem with this method, as I highly doubt it represents the response of the entire organism. When where the incubations done, at the end of the experiment? How many replicates were used?

L176: This was also done at the end of the 9 d?

Results: I would favour indicating the actual p-values rather than < 0.05 or >0.05

L-314-315: Any reason why the algae would do that? If they have more carbon available why would they reduce their photosynthesis? It doesn't make much sense from an organismal point of view.

L 331-332: Could this be due to pH rather than carbon?

L344-345: CCM activity has often been linked to the light level. Could it explain some of these results?

As explained before, I think that an additional figure to summarize all of those results (and mostly the link between each other) would be highly valuable.

L392-393: Could the seaweed culture also be affected by those limitations?

---

## Referee Comment (RC3) · Anonymous Referee #4 · 2 Apr 2018

This manuscript details the results of a classical pCO2 x Nutrients experiment with seaweeds. In that respect its novelty relays in the distinction between N and P limitation, while most of the phenomena concerning pCO2 x N has been described before in Ulva sp. (eg. Gordillo et al. 2001 Planta and Gordillo et al. 2003 Planta).

Main comments

A major concern is about net photosynthesis. As it is measured (O2 evolution), changes can derive either from photosynthesis or from respiration. Since respiration of seaweeds is commonly affected by pCO2 (Iñiguez et al. 2015 Polar Biol.; Iñiguez et al. 2016 Mar Biol) even in Ulva (e.g. Gordillo et al. 2003 Planta) and also by nutrients,

authors must show respiration rates along with the net or gross photosynthesis. Otherwise, not much can be said about the effect of pCO2 and nutrients on photosynthetic O2 evolution.

Line 304. The 'pigment economy' phenomenon occurring in algae at high pCO2 was first described in Gordillo et al. (1999 J appl. Phycol) and described for Ulva using exactly the same name by Gordillo et al. 2003 (Planta), so credit must be given to those authors.

Minor comments Methods Incubation setup needs more detail. What type of recipient was used for adult thalli? At what density? Was the bubbling enough to make them move or were they settling on the bottom? Incubation light need more detail. What source of light was used (fluorescent tubes of daylight type?). Also how was the irradiance measured? (type of sensor, air or underwater?, lambda range?PAR?)

53. 'also' instead of 'only' 148-150. Sentence is nonsensical, please rephrase. 164. Units needed (nm)

Tables 4 to 7 can be combined and look like table 2, so the information is not scattered.

Fig.2. The horizontal bar means significant differences between LC and HC, but that is hard to believe for some of the treatment at least like LNHP in (a), and HNHP and LNHP in (b). Please check your post-hoc comparisons. It is also highly convenient you mention the number of replicates (n) in the figure legends.

---

## Author Response (AR1)

**Response to reviewers' comments**

**Anonymous Referee #2**

Comments: The authors investigated the combined effects of ocean acidification and nutrient limitation on physiological performances, including growth, photosynthetic oxygen evolution, PSII fluorescence parameters, nitrogen assimilation, in a green tide alga, and found that ocean acidification did not affect growth and photosynthesis under the nutrient replete condition but reduced them when nutrient was limited. Nitrogen assimilation was stimulated by ocean acidification when nutrient was replete. The experiments were reasonably performed and the results were clearly presented. This study is of interest, indicating the interactive effects of global and local stressors on a green tide alga. But there are still some points to be revised before it could be published in Biogeosciences.

Response: We appreciate these comments very much and revised the manuscript based on the reviewer's comments.

Special revisions

1. Why were different cultivation periods used for sporeling and adult thalli? Are these periods enough for algal acclimation to ocean acidification?

Response: The cultures had been finished before the thalli became reproductive as the aim of this study focused on the growth and photosynthesis. Different cultivation periods were used because the periods were different for sporeling and adult to become reproductive. These cultivation periods are enough for *Ulva linza*'s acclimation to ocean acidification (Eggert, 2012; Gao et al., 2016). This information has been added to lines 147-151.

Eggert A. Seaweed responses to temperature. In: Wiencke C, Bischof K, editors. Seaweed biology. Berlin: Springer; 2012. pp.47-66.

Gao G, Liu Y, Li X, et al. An ocean acidification acclimatised green tide alga is robust to changes of seawater carbon chemistry but vulnerable to light stress. PloS one, 2016, 11(12): e0169040.

2. Please clarify the culture density used in this study and to what extent pH fluctuated during the culture period. How to maintain a stable pH in the cultures?

Response: The culture density was less than 0.1 g  $L^{-1}$  and the pH fluctuation was less than 0.03 units. Low culture density and aeration with ambient and CO2-enriched air contributed to the stable pH in the cultures. This information has been added to lines 142-145.

3. Why was the light density of 300 photons m-2 s-1 used for the cultures since lower levels were used for the previous studies as mentioned in the text. Is the one used in this study close to ambient sunlight?

Response: The samples were collected in March 2017 and the light density of 300 photons  $m^{-2} s^{-1}$  used for the cultures was close to the ambient light level at the sample collecting site. This information has been added to lines 134-136.

Minor revisions
Line 113 change µmol to µmol photons m-2 s-1 Response: Corrected.
Line 123 add a space after 106.1 Response: Corrected.
Line 156 change weight to mass Response: Corrected.
Line 329 delete activity and be consistent for using NRA or NR activity throughout the text. Response: Corrected.
Figure 3 change FW to FM in Y axes legend

Response: Corrected.

Figure 7 I doubt there is a significant difference between HC and LC for the treatment of HNLP

Response: No, there is no significant difference between HC and LC for the treatment of HNLP. We apologize for this mistake and it has been corrected.

**Anonymous Referee #3**

This paper reports results from an interesting study aiming to test the effects of ocean acidification and nutrients limitation on Ulva. The study is pretty straightforward: adult and juvenile algae were exposed to different conditions of CO2 and nutrients and their physiological response was investigated. While this study is rather "classical", the originality comes from the use of nutrient limitation, while most studies have used so far nutrients addition. The results are rather interesting and demonstrate that the interaction between pCO2 and nutrient limitations are not straightforward. I find the discussion a bit complex and hard to read given the quantity of physiological parameters discussed. It might be worth considering adding a figure that would summarize all the results. Maybe a schematic representing the physiological impact of nutrients and carbon could be added. I have listed below some specific comments.

Response: We appreciate these comments very much and a schematic figure (Fig. 8) has been added to summarize all the results.

Abstract: indicate the duration of the experiment

Response: it has been clarified to "We cultured *Ulva linza* for 9-16 days" at line 6.

L55: Wrong reference for Cornwall et al. 2017, they looked at coralline algae not phytoplankton.

Response: We are grateful for this comment and this reference has been removed. L63-64: reformulate this sentence

Response: It has been revised to "By analyzing the literatures, it is found that life stage can affect the effects of ocean acidification on growth of *Ulva* species" at lines 63-65.

L119: "LCHNHP" is a bit hard to read/understand but I guess it's not really used later on.

Response: It has been revised to "The treatment of lower pCO2, higher nitrate and higher phosphate (LCHNHP) was set as the control." at lines 121-122. L130: How does this light level compare to in situ?

Response: The samples were collected in March 2017 and the light density of 300 photons  $m^{-2} s^{-1}$  used for the cultures was close to the ambient light level in situ. This information has been added to lines 134-136.

L132: What was the size of the tanks? Did you use any pumps, etc , to create water motion? This is critical as it could affect the capacity of the organisms to uptake nutrients.

Response: The thalli were grown in 1-L balloon flasks containing 900 mL of media. The cultures were bubbled with ambient or  $CO_2$ -enriched air at a rate of 300 mL min-1 to make the thalli roll up and down. Please see lines 140-141. L133: Any reason to have chosen these durations? 9 days is rather short.

Response: The cultures had been finished before the thalli became reproductive as the aim of this study focused on the growth and photosynthesis. Different cultivation periods were used because the periods were different for sporeling and adult to become reproductive. These cultivation periods are enough for *Ulva linza*'s acclimation to ocean acidification (Eggert, 2012; Gao et al., 2016). This information has been added to lines 147-151.

L156: What were those fragments? Just a piece of algae? I always have problem with this method, as I highly doubt it represents the response of the entire organism. When where the incubations done, at the end of the experiment? How many replicates were used?

Response: The text has been specified to "Algal individuals were cut into 1-cm-long segments with a scissor. Approximately 0.02 g segments were randomly selected and transferred to the oxygen electrode cuvette with 8 ml of media from the culture flask." at lines 175-178 and " The following parameters were measured at the end of the culture periods for each flask under each treatment." at lines 151-152. L176: This was also done at the end of the 9 d?

Response: Yes.

Results: I would favour indicating the actual p-values rather than < 0.05 or >0.05

Response: We have used the actual P-values for most cases, with P < 0.001 for those where actual P-values were less than 0.001. Meanwhile, we hope we can keep P < 0.05 or > 0.05 for some cases where there are too many comparisons in one sentence.

L-314-315: Any reason why the algae would do that? If they have more carbon available why would they reduce their photosynthesis? It doesn't make much sense from an organismal point of view.

Response: We appreciate these comments. The explanation has been specified to "Meanwhile, the saved energy due to down-regulation of CCMs in thalli grown under HC combined with higher light density used in this study may depress PSII activity and thus reduce net photosynthetic rate (Gao et al., 2012)." at lines 341-344. L 331-332: Could this be due to pH rather than carbon?

Response: Yes, there is possibility that the change of NRA was due to pH. The following information has been added to the text "Meanwhile, the change of NRA under different  $pCO_2$  levels might be also caused by varying pH as pH could affect NRA in seaweeds (Lopes et al., 1997)" at lines 369-370.

Lopes P F, Oliveira M C, Colepicolo P. Diurnal fluctuation of nitrate reductase activity in the marine red alga *Gracilaria tenuistipitata* (Rhodophyta). Journal of Phycology, 1997, 33: 225-231.

L344-345: CCM activity has often been linked to the light level. Could it explain some of these results?

Response: Yes, there are connections among CCM activity,  $CO_2$  and light. The related discussion has been added to the text and it reads " The potential reason is that the saved energy from down-regulated CCMs under higher  $CO_2$  levels could be used for growth at lower light levels but could inhibit PSII activity and thus growth at higher light levels" at lines 388-391.

As explained before, I think that an additional figure to summarize all of those results (and mostly the link between each other) would be highly valuable.

Response: It has been done. Please see section 4.4 for details.

L392-393: Could the seaweed culture also be affected by those limitations? Response: The text has been revised to "This may hinder the occurrence of green tides and *Ulva* cultivation in future ocean." at lines 436-437.

**Anonymous Referee #4**

This manuscript details the results of a classical pCO2 x Nutrients experiment with seaweeds. In that respect its novelty relays in the distinction between N and P limitation, while most of the phenomena concerning pCO2 x N has been described before in Ulva sp. (eg. Gordillo et al. 2001 Planta and Gordillo et al. 2003 Planta).

Response: We agree with these comments. Gordillo et al (2001, 2003) did excellent work on the interaction of  $CO_2$  and N. Another novel point of our study is that we used diluted natural seawater as nutrient limiting condition rather than natural seawater to mimic the situation in seaweed cultivation areas. Main comments

A major concern is about net photosynthesis. As it is measured (O2 evolution), changes can derive either from photosynthesis or from respiration. Since respiration of seaweeds is commonly affected by pCO2 (Iñiguez et al. 2015 Polar Biol.; Iñiguez et al. 2016 Mar Biol) even in Ulva (e.g. Gordillo et al. 2003 Planta) and also by nutrients, authors must show respiration rates along with the net or gross photosynthesis. Otherwise, not much can be said about the effect of pCO2 and nutrients on photosynthetic O2 evolution.

Response: We totally agree with these comments. We measured dark respiration rate, but did not represent it as neither  $pCO_2$  nor nutrient affected it, indicating that changes of  $O_2$  evolution derived from photosynthesis rather than respiration. The data of dark respiration have been added to the text and also been discussed. Please see lines 259-260 and 354-360.

Line 304. The 'pigment economy' phenomenon occurring in algae at high pCO2 was

first described in Gordillo et al. (1999 J appl. Phycol) and described for Ulva using exactly the same name by Gordillo et al. 2003 (Planta), so credit must be given to those authors.

Response: We agree with these comments and the text has been corrected to "This phenomenon of `pigment economy' has also been found in the previous studies regarding *Ulva* species (Gordillo et al., 2003; Gao et al., 2016)." at lines 330-332. Minor comments Methods Incubation setup needs more detail. What type of recipient was used for adult thalli? At what density? Was the bubbling enough to make them move or were they settling on the bottom? Incubation light need more detail. What source of light was used (fluorescent tubes of daylight type?). Also how was the irradiance measured? (type of sensor, air or underwater?, lambda range?PAR?)

Response: We appreciate these comments. The thalli were grown in 1-L balloon flasks containing 900 mL of media with the density less than 0.1 g L-1. The cultures were bubbled with ambient or CO2-enriched air at a rate of 300 mL min-1 to make the thalli roll up and down. Daylight fluorescent tubes (21W, Philips) were used and light density was measured by a Quantum Scalar Laboratory (QSL) radiometer (QSL-2100, Biospherical Instruments, Inc., USA) that detects photosynthetically active radiation (400-700 nm). Please see section 2.1.

53. 'also' instead of 'only'

Response: Corrected.

148-150. Sentence is nonsensical, please rephrase.

Response: It has been corrected to "The measuring light was 0.01 µmol photons  $m^{-2} s^{-1}$  and actinic light was set as the same as the growth light (300 µmol photons  $m^{-2} s^{-1}$ )" at lines 167-169.

164. Units needed (nm)

Response: Corrected.

Tables 4 to 7 can be combined and look like table 2, so the information is not scattered.

Response: Tables 4 to 7 has been combined into a table, termed table 4. Fig.2. The horizontal bar means significant differences between LC and HC, but that is hard to believe for some of the treatment at least like LNHP in (a), and HNHP and LNHP in (b). Please check your post-hoc comparisons. It is also highly convenient you mention the number of replicates (n) in the figure legends.

Response: The real indication of horizontal bars is that longer bars represent insignificant differences and shorter bars represent significant differences. We have realized that it is a little confusing. We have removed the longer horizontal bars to make it clear.

**Comments from Dinghui Zou**

The authors investigated the combined effects of ocean acidification and nutrient limitation on physiological performances, including growth, photosynthetic oxygen evolution, PSII fluorescence parameters, nitrogen assimilation, in a green tide alga, and found that ocean acidification did not affect growth and photosynthesis under the nutrient replete condition but reduced them when nutrient was limited. Nitrogen assimilation was stimulated by ocean acidification when nutrient was replete. The experiments were reasonably performed and the results were clearly presented. This study is of interest, indicating the interactive effects of global and local stressors on a green tide alga. But there are still some points to be revised before it could be published in Biogeosciences.

Response: We appreciate these comments very much and revised the manuscript based on the reviewer's comments.

Special revisions

1. Why were different cultivation periods used for sporeling and adult thalli? Are these periods enough for algal acclimation to ocean acidification?

Response: The cultures had been finished before the thalli became reproductive as the aim of this study focused on the growth and photosynthesis. Different cultivation periods were used because the periods were different for sporeling and adult to become reproductive. These cultivation periods are enough for *Ulva linza*'s acclimation to ocean acidification (Eggert, 2012; Gao et al., 2016). This information has been added to lines 147-151.

Eggert A. Seaweed responses to temperature. In: Wiencke C, Bischof K, editors. Seaweed biology. Berlin: Springer; 2012. pp.47-66.

Gao G, Liu Y, Li X, et al. An ocean acidification acclimatised green tide alga is robust to changes of seawater carbon chemistry but vulnerable to light stress. PloS one, 2016, 11(12): e0169040.

2. Please clarify the culture density used in this study and to what extent

pH fluctuated during the culture period. How to maintain a stable pH in the cultures? Response: The culture density was less than 0.1 g L-1 and the pH fluctuation was less than 0.03 units. Low culture density and aeration with ambient and CO2-enriched air contributed to the stable pH in the cultures. This information has been added to lines 142-145.

3. Why was the light density of 300 photons m-2 s-1 used for the cultures since lower levels were used for the previous studies as mentioned in the text. Is the one used in this study close to ambient sunlight?

Response: The samples were collected in March 2017 and the light density of 300 photons  $m^{-2} s^{-1}$  used for the cultures was close to the ambient light level at the sample collecting site. This information has been added to lines 134-136.

Minor revisions

Line 113 change µmol to µmol photons m-2 s-1

Response: Corrected.

Line 123 add a space after 106.1 Response: Corrected.

Line 156 change weight to mass

Response: Corrected.

Line 329 delete activity and be consistent for using NRA or NR activity throughout the text.

Response: Corrected.

Figure 3 change FW to FM in Y axes legend

Response: Corrected.

Figure 7 I doubt there is a significant difference between HC and LC for the treatment of HNLP

Response: No, there is no significant difference between HC and LC for the treatment of HNLP. We apologize for this mistake and it has been corrected.

[revised manuscript text omitted]

interactive effect of pCO2 and nutrient, df means degree of freedom, F means the value of F statistic, and Sig. means p-value.

Table 4. Two-way analysis of variance for the effects of pCO2 and nutrient on physiological parameters of U. linza. pCO2\*nutrient means the

| Table 4. Two-way analysis of variance for the effects of pCO 2 and nutrient on relative |
|----------------------------------------------------------------------------------------------------|
| growth rate of U. linza. $pCO_2$ *nutrient means the interactive effect of $pCO_2$ and             |
| nutrient, df means degree of freedom, F means the value of F statistic, and Sig. means             |
| p -value.                                                                                   |

|                             | Growth of young U. linza |                      |                      | Growth of adult U. linza |                    |                      |
|-----------------------------|--------------------------|----------------------|----------------------|--------------------------|--------------------|----------------------|
| Source                      | đf                       | F                    | <del>Sig.</del>      | df                       | Ŧ                  | <del>Sig.</del>      |
| <del>pCO</del> 2 | 4                        | <del>115.297</del>   | <del><0.001</del> | 4                        | <del>20.039</del>  | <del><0.001</del> |
| Nutrient                    | 3                        | <del>12678.566</del> | <del><0.001</del> | 3                        | <del>307.073</del> | <del><0.001</del> |
| pCO2*nutrient               | 3                        | <del>22.905</del>    | <del><0.001</del> | 3                        | <del>1.723</del>   | <del>0.011</del>     |
| Error                       | <del>16</del>            |                      |                      | <del>16</del>            |                    |                      |

**Table 5.** Two way analysis of variance for the effects of  $pCO_2$  and nutrient on net photosynthetic rate and rETR of *U. linza.*  $pCO_2$ \*nutrient means the interactive effect of  $pCO_2$  and nutrient, df means degree of freedom, F means the value of F statistic, and Sig. means *p* value.

|                             | Net photosynthetic rate |                    |                      | <del>rETR</del> |                    |                      |
|-----------------------------|-------------------------|--------------------|----------------------|-----------------|--------------------|----------------------|
| Source                      | đf                      | F                  | Sig.                 | <del>df</del>   | F                  | Sig.                 |
| <del>pCO</del> 2 | 4                       | <del>35.096</del>  | <del><0.001</del> | 4               | <del>14.592</del>  | <del>0.002</del>     |
| Nutrient                    | <del>3</del>            | <del>493.992</del> | <del><0.001</del> | 3               | <del>135.690</del> | <del><0.001</del> |
| <del>pCO₂*nutrient</del>    | 3                       | <del>2.619</del>   | <del>0.087</del>     | 3               | <del>5.023</del>   | <del>0.012</del>     |
| Error                       | <del>16</del>           |                    |                      | <del>16</del>   |                    |                      |
|                             |                         |                    |                      |                 |                    |                      |

**Table 6.** Two-way analysis of variance for the effects of  $pCO_2$ -and nutrient on content of Chl *a* and Chl *b* in *U. linza.*  $pCO_2$ \*nutrient means the interactive effect of  $pCO_2$ and nutrient, df means degree of freedom, F means the value of F statistic, and Sig. means *p*-value.

|                             | Chl-a         |                    |                      | Chl-b         |                    |                      |
|-----------------------------|---------------|--------------------|----------------------|---------------|--------------------|----------------------|
| Source                      | đf            | F                  | <del>Sig.</del>      | df            | F                  | <del>Sig.</del>      |
| <del>pCO</del> 2 | 4             | <del>85.900</del>  | <del><0.001</del> | 4             | <del>71.600</del>  | <del><0.001</del> |
| Nutrient                    | <del>3</del>  | <del>217.334</del> | <del><0.001</del> | <del>3</del>  | <del>104.483</del> | <del><0.001</del> |
| <del>pCO₂*nutrient</del>    | 3             | <del>2.440</del>   | <del>0.102</del>     | 3             | <del>2.005</del>   | <del>0.154</del>     |
| Error                       | <del>16</del> |                    |                      | <del>16</del> |                    |                      |

**Table 7.** Two-way analysis of variance for the effects of  $pCO_2$  and nutrient on nitrate reductase activity and soluble protein of *U. linza*.  $pCO_2$ \*nutrient means the interactive effect of  $pCO_2$  and nutrient, df means degree of freedom, F means the value of F statistic, and Sig. means *p*-value.

|                             | Nitrate reductase activity |                    |                      | Soluble protein |                    |                      |
|-----------------------------|----------------------------|--------------------|----------------------|-----------------|--------------------|----------------------|
| Source                      | đf                         | F                  | <del>Sig.</del>      | <del>df</del>   | F                  | Sig.                 |
| <del>pCO</del> 2 | 4                          | <del>38.271</del>  | <del><0.001</del> | 4               | <del>30.212</del>  | <del><0.001</del> |
| Nutrient                    | 3                          | <del>100.487</del> | <del><0.001</del> | 3               | <del>106.523</del> | <del><0.001</del> |
| pCO2*nutrient               | <del>3</del>               | <del>6.246</del>   | <del>0.005</del>     | <del>3</del>    | <del>11.295</del>  | <del><0.001</del> |
| Error                       | <del>16</del>              |                    |                      | <del>16</del>   |                    |                      |

**Figure legends**

[revised manuscript text omitted]

Fig. 1